# The Establishment of Expanded Newborn Screening in Rural Areas of a Developing Country: A Model from Health Regions 7 and 8 in Thailand

**DOI:** 10.3390/ijns11020026

**Published:** 2025-04-12

**Authors:** Khunton Wichajarn, Nopporn Sawatjui, Prinya Prasongdee, Amrin Panklin, Kanda Sornkayasit, Natchita Chungkanchana, Supharada Tessiri, Preawwalee Wintachai, Sumalai Dechyotin, Chalanda Pasomboon, Jilawaporn Ratanapontee, Sureerat Thanakitsuwan, Aree Rattanathongkom

**Affiliations:** 1Division of Medical Genetics, Department of Pediatrics, Faculty of Medicine, Khon Kaen University, Khon Kaen 40002, Thailand; areera@kku.ac.th; 2Center of Excellence in Precision Medicine, Srinagarind Hospital, Faculty of Medicine, Khon Kaen University, Khon Kaen 40002, Thailand; kandaso@kku.ac.th (K.S.); jilara@kku.ac.th (J.R.); suri.khwanmanee@gmail.com (S.T.); 3Clinical Laboratory Section, Srinagarind Hospital, Faculty of Medicine, Khon Kaen University, Khon Kaen 40002, Thailand; pparin@kku.ac.th (P.P.); amrinpa@kkumail.com (A.P.); prew.natchita@gmail.com (N.C.); supharada.tsr@gmail.com (S.T.); preawi@kku.ac.th (P.W.); sumalai@kku.ac.th (S.D.); chalanda.pasomboon@gmail.com (C.P.); 4Srinagarind Excellence Laboratory, Faculty of Medicine, Khon Kaen University, Khon Kaen 40002, Thailand

**Keywords:** expanded newborn screening, inborn errors of metabolism, congenital hypothyroidism, rural healthcare, Thailand

## Abstract

Expanded newborn screening (NBS) programs are essential for early detection and treatment. This study highlights the implementation of an expanded NBS program for inborn errors of metabolism (IEMs) and congenital hypothyroidism (CH) in rural Thailand, focusing on Health Regions 7 and 8 as a model for resource-limited settings. Using the KKU-IEM web-based platform, the program streamlined workflows, integrating logistics, real-time sample tracking, and electronic data management. Regular training sessions, continuous feedback, and systematic monitoring improved outcomes. Starting from October 2022, the program covered 98.6% of 123,692 live births, identifying 101 CH cases (1 in 1208 live births) and 20 IEM cases (1 in 6100 live births). The CH incidence was slightly higher than Thailand’s national average, while the IEM incidence was double that found in a previous Bangkok pilot study. Six cases highlighted maternal conditions affecting outcomes. Process improvements reduced the average reporting time from 9.13 days in 2023 to 8.4 days in 2024, with a 19% reduction in Bueng Kan Province. Efficiencies were driven by electronic ordering, real-time tracking, and stakeholder collaboration. This program demonstrates a scalable model for rural settings, emphasizing technology integration, collaboration, and quality control. Future efforts should refine diagnostics, expand disease coverage, and enhance long-term outcomes.

## 1. Introduction

Inborn errors of metabolism (IEMs) represent a group of inherited disorders characterized by defects in various biochemical metabolic pathways. Many of these conditions manifest clinically during the early neonatal period, with signs and symptoms often resembling neonatal sepsis, which makes accurate diagnosis challenging. Failure to diagnose these conditions promptly can lead to poor outcomes, including increased infant mortality rates and long-term disabilities.

Newborn screening (NBS) for IEMs began with phenylketonuria (PKU), which was first reported by Dr. Robert Guthrie in 1961 [1]. Since then, screening programs have expanded to include other IEMs and conditions. Today, expanded NBS for IEMs via tandem mass spectrometry (MS/MS) is a powerful tool for early diagnosis and has been widely implemented in many developed countries.

In Thailand, NBS was first piloted in 1993 and became mandatory nationwide in 1996, covering two conditions: congenital hypothyroidism (CH) and PKU [2]. In 2014, Liammongkolkul et al. [3] piloted a study at Siriraj Hospital, introducing expanded NBS for 40 IEMs. This project screened 99,234 neonates born in 15 public hospitals in Bangkok and reported an incidence of 1 in 6616 births (or 1 in 12,404 births if cases due to affected mothers were excluded).

In 2022, Thailand’s National Health Security Office (NHSO) approved a benefit package covering the diagnosis and treatment of 24 rare IEMs, providing universal health coverage for Thai citizens. Subsequently, seven rare disease (RD) centers were established, with six located in the Bangkok Metropolitan Region and one at Srinagarind Hospital in Khon Kaen, Northeastern Thailand. The Health Intervention and Technology Assessment Program (HITAP) conducted an economic evaluation and feasibility study of expanded NBS using MS/MS. The results showed that early treatment, prior to the onset of symptoms, is more cost-effective compared to late treatment and recommended that the NHSO include expanded NBS with MS/MS technology under the Universal Coverage Scheme [4].

Following the HITAP’s recommendations, the expanded NBS program for all Thai neonates was launched in October 2022. A total of 10 NBS centers were established, covering all 13 health regions nationwide. Srinagarind Hospital serves as both an RD center and an NBS center, overseeing Health Regions (HRs) 7 and 8 in Northeastern Thailand, responsible for more than 50,000 births annually across 11 provinces.

## 2. Materials and Methods

### 2.1. System Development and Validation

The Srinagarind Excellence Center Lab (SEL), which includes the Genomics and Precision Medicine Laboratory, in collaboration with the Center of Excellence in Precision Medicine and the Division of Medical Genetics in the Department of Pediatrics at the Faculty of Medicine, Srinagarind Hospital, Khon Kaen University, serves as the main unit responsible for the NBS service. A GSP Neonatal hTSH kit is used for CH screening via the DELFIA assay, while a NeoBase 2 Non-derivatized MSMS kit is used for IEM screening via MS/MS. The target disorders for IEMs are shown in Table 1 [4].

Over 2000 blood spot samples were used for validation and to set up the initial cutoff values for TSH and MS/MS analysis. Simultaneously with the validation process, the workflow system was developed based on a lean concept to minimize human error. The online platform was developed to facilitate the process from registration, electronic ordering, logistics, interpretation, and reporting. The workflow system is shown in Figure 1.

### 2.2. Software and Logistics Preparation

Since NBS requires collaboration with many hospitals, a software system was developed to facilitate data exchange and reporting. The KKU-IEM web-based online platform (https://www.kku-iem.com) was created to record newborn and maternal data, along with sample information, and to track sample status in real time from data entry to sample shipment, laboratory receipt, and result reporting. The system generates lab numbers and barcodes for each sample, streamlining laboratory operations. Individual and hospital-specific reports can be printed, with restricted access via username and password for each coordinator in each hospital. Data retrieval and export for analysis or submission to the NHSO is also available. In terms of logistics, the service partnered with Thailand Post Distribution (TPD) to collect samples from all hospitals daily. Once the KKU-IEM platform records a shipment, Thailand Post collects and delivers the samples to our NBS center.

### 2.3. Training of Networks

Nurse, doctor, and medical technologist coordinators from each network hospital were assigned to collaborate with the NBS lab in Srinagarind Hospital. Prior to the launch of the services, workshops were conducted with the hospital network in HR 7 (four provinces including Kalasin, Khon Kaen, Maha Sarakham, and Roi-Et) and HR 8 (seven provinces including Bueng Kan, Loei, Nakhon Phanom, Nong Bua Lamphu, Nong Khai, Sakon Nakhon, and Udon Thani). These sessions provided training on sample collection, electronic registration through the KKU-IEM platform, sample tracking, recalling infants with positive results, evaluating the affected infants, and consulting the geneticist at Srinagarind Hospital (RD center) for further management in accordance with the established guidelines. To ensure smooth operations, online meetings and site visits were organized to gather stakeholder feedback, address issues, and implement system improvements.

All newborns are recommended to undergo NBS with blood collection at 48–72 h of age. For preterm or low-birth-weight infants, secondary NBS is recommended at 2–3 weeks of age. If the results are positive, the nurse coordinator or medical geneticist will inform the network hospital coordinator to recall the patient for evaluation.

According to the consensus of the committee in the working group of the rare diseases policy under the NHSO of Thailand, the screened diseases were categorized into two groups based on the urgency of recalling babies after they received positive screening results [4] (Table 1). These categories are as follows:Very Urgent: Infants must be followed up with for evaluation within 24 h of receiving a positive result.Urgent: Infants must be followed up with for evaluation within 48 h of receiving a positive result.

To facilitate effective communication, multiple channels were established between network hospitals and the NBS center. A dedicated a LINE Official Account, NB Screening KKU, was created to handle inquiries, blood filter paper requests, and service updates. The account also features a menu with detailed program instructions and guidelines for proper blood sample handling.

Additionally, direct contact numbers for the pediatric geneticist and the NBS laboratory were provided, along with email communication channels to ensure accessibility. Comprehensive screening protocols and follow-up procedures for managing abnormal results were also established to streamline the process and ensure timely interventions.

### 2.4. Quality Control

The cut-off was re-evaluated 6 and 12 months after implementation with 10,000 and 50,000 samples, respectively. An inter-lab comparison project was established among three NBS centers: Srinagarind Hospital, Queen Sirikit National Institute of Child Health, and Siriraj Hospital. Furthermore, the center joined the Centers for Disease Control and Prevention (CDC) for proficiency testing and external quality control to ensure the quality of laboratory analyses. The lab also received international accreditation for ISO 15189 and ISO 15190 standards (currently accredited).

### 2.5. Re-Evaluation

We regularly set up internal evaluations every quarter of the year in addition to a comprehensive annual performance report. The report outlines NBS results from the fiscal year, reviews operational practices, reflects on issues encountered during implementation, and provides a platform for knowledge exchange with the hospital network. It also includes an opportunity for suggestions, conducting a satisfaction survey, and exploring opportunities for improvement, particularly in reducing the time it takes to report results so infants can be diagnosed faster.

An online video clip about the NBS system at Srinagarind Hospital was created to communicate the importance and procedures of NBS to the public. Continuous improvements were made to the KKU-IEM web program, and the results report form was revised to be more user-friendly, making the process easier for both service units and the screening laboratory. Knowledge management and sharing among the other NBS centers was established through annual conferences.

## 3. Results

The NBS network comprised 169 hospitals (79 in HR 7 and 90 in HR 8). All network hospitals utilized the KKU-IEM web-based platform for electronic online ordering. From October 2022 to September 2024, there were 123,692 live births in these regions (12.4% of the whole country), of which 122,004 newborns (98.6%) underwent screening. Among them, 17,528 were preterm (12,164) and/or low-birth-weight (13,179) newborns. Secondary NBS was performed on 11,138 infants (9.1%).

Abnormal screening results included 287 newborns who tested positive for CH and 529 newborns who tested positive for IEMs. The recall rates were 1 in 425.1 for CH and 1 in 230.6 for IEMs. The successful follow-up rates for confirmatory testing were 99.0% for newborns with positive CH screening results and 96.0% for those with positive IEM results. A total of 101 neonates were confirmed to have CH based on abnormal thyroid-stimulating hormone (TSH) or free thyroxine (T4) in serum, with an incidence of 1 in 1208 live births. Additionally, 20 neonates were diagnosed with IEMs, corresponding to an incidence of 1 in 6100 live births. Maternal conditions (maternal carnitine uptake defect and maternal 3-methylcrotonyl-CoA carboxylase deficiency) affecting NBS results were identified in six neonates with positive IEM results. These findings are summarized in Table 2. The recall rates observed in this study were 1 in 425.1 for CH, with a positive predictive value (PPV) of 35.2%, and 1 in 221.0 for IEMs, with a PPV of 3.8%. The IEMs found in the NBS program are demonstrated in Figure 2.

Key performance indicators (KPIs) set by the Department of Medical Science, Ministry of Public Health, specify that laboratory processing should be completed within 3–5 days, and the birth-to-report duration should not exceed 14 days. The NBS laboratory at Srinagarind Hospital exceeded these benchmarks, achieving an average laboratory processing time of 1.4 days. The process outcomes showed improvement, with the average birth-to-report duration reduced from 9.13 days in 2023 to 8.4 days in 2024. The birth-to-report duration improved across all 11 provinces from 2023 to 2024, with Bueng Kan Province achieving the highest reduction at 19.1%. This improvement followed the presentation of performance reports at a conference and led to subsequent workflow adjustments in collaboration with network hospitals.

## 4. Discussion

NBS is an important tool for advancing the United Nations Sustainable Development Goal (SDG) 3: Good Health and Well-Being. By enabling the early detection of congenital and metabolic disorders, NBS significantly improves health outcomes through timely intervention and treatment. Many countries have developed tailored NBS programs, each designed to address their specific public health needs and target disorders. The expanded NBS program in Thailand has transitioned to a decentralized model, with Srinagarind Hospital uniquely positioned as one of ten NBS centers. Before decentralization, the NBS program was primarily managed by the Department of Medical Science, Ministry of Public Health, located in the Bangkok Metropolitan Region. Srinagarind Hospital is strategically located and optimally placed to serve HRs 7 and 8 in Northeastern Thailand. Furthermore, it is the only RD center located outside the Bangkok metropolitan area, providing a distinct advantage in facilitating collaboration with network hospitals. Srinagarind Hospital functions as a one-stop service center for screening, confirmatory testing, and treatment, making it a reliable hub for network hospitals. Its integrated services ensure timely and expert support, including consultations for positive NBS results, initial management recommendations, and rapid patient transfers if necessary.

Its strategic location significantly reduces logistical challenges for the region. For instance, the farthest province in its catchment area, Bueng Kan in HR 8, is approximately 270 km (a little over a 4 h drive) from Srinagarind Hospital in Khon Kaen. If a baby born in Bueng Kan receives a positive result, the referral to Srinagarind Hospital is far more feasible compared to a referral to Bangkok, which is over 720 km away—a journey that would take more than 11 h by car. This proximity ensures more timely access to confirmatory testing and specialized care, which is critical for conditions requiring immediate intervention.

This disparity highlights a strategic gap in the national NBS program. The Srinagarind model demonstrates the critical importance of aligning NBS and RD centers within the same catchment area. This integration shortens processes, ensures timely care, and can significantly improve outcomes for time-sensitive conditions such as IEMs, where delays in diagnosis and treatment can be life-threatening. This model serves as a model for optimizing NBS programs in other regions.

The expanded NBS program in HRs 7 and 8 of Thailand has proven to be a valuable tool for implementing these initiatives in rural areas of developing countries. This program achieved a screening coverage rate of 98.6%, which is comparable to or exceeds rates reported in many developing countries in Asia [5]. However, several factors may contribute to the remaining gap up to 100% coverage, even after implementing the service across both regions. One key factor is the flexibility of birth registration, which can be completed within 15 days after birth. This flexibility contrasts with NBS, which is conducted at 48 h of age. Additionally, birth registration can be carried out at any civil registration office, regardless of the baby’s place of birth. Population movement also plays a role, as pregnant women may relocate due to work or be referred to tertiary care hospitals in different areas for delivery. After giving birth, these mothers may register their newborns at a registration office in a different location from the birth hospital. These factors can lead to slight discrepancies between the number of screenings conducted in each region and the number of births reported. To address this, compiling data from all NBS centers nationwide is essential for providing the most accurate representation of screening coverage in Thailand.

Globally, expanded NBS programs in developed countries often report confirmatory rates of over 95% for positive cases, which is similar to the results achieved for both CH and IEMs in this study. However, the 96.0% follow-up rate for IEMs was lower than that for CH, indicating room for improvement in rural Thailand. Factors such as healthcare access disparities and limited public awareness of metabolic disorders might contribute to this difference.

The recall rates observed in this study were 1 in 425.1 for CH, with a positive predictive value (PPV) of 0.35, and 1 in 221.0 for IEMs, with a PPV of 0.05. While the recall rate and PPV for CH fall within acceptable ranges, the recall rate for IEMs indicates a higher burden. This disparity may be attributed to the broader spectrum of disorders included in the screening panel, as well as several potential interferences such as extremely preterm infants, a low birth weight, birth asphyxia, and the use of total parenteral nutrition. Additionally, the lower threshold for positive findings in our program may contribute to the higher recall rate for IEMs. These findings highlight the need for the ongoing evaluation and refinement of screening protocols to balance sensitivity and specificity, particularly for IEMs, and to minimize unnecessary recalls while maintaining the effective detection of true positives. Enhanced data collection and analysis regarding the causes of false positives could further optimize the screening process and reduce the burden on families and healthcare systems.

The program identified 101 confirmed cases of CH and 20 confirmed cases of IEMs among the 122,004 newborns screened. The incidence rate of CH was 1 in 1208 live births, which is slightly higher than the global average of 1 in 2000–4000 births [6] and Thailand’s average of 1 in 1708 births [7]. Further analysis found that Udon Thani Province has the highest incidence at 1 in 816 births. This finding suggests potential regional or ethnic predispositions. Another factor contributing to the higher CH incidence in this study compared to earlier reports in Thailand may be related to the NBS methodology. The current NBS program for CH screening may include transient CH cases, accounting for approximately 29% of CH diagnoses [8]. The most important factor of transient CH is iodine deficiency. A study in Thailand in 2008 found that many provinces in our catchment area had a moderate degree of iodine deficiency, including Udon Thani [9]. This might reflect a higher incidence of transient CH. This is likely due to the limited availability of long-term systematic follow-up data for infants diagnosed with CH, which would be necessary to distinguish transient cases from permanent CH. These results emphasize the need for comprehensive follow-up systems to ensure accurate classification and better understanding of CH incidence trends.

The incidence of IEMs in this study was 1 in 6100 live births, which is twice as high as the rate reported in previous pilot studies conducted in Bangkok, Thailand [3]. Further investigations revealed regional genetic variations that contribute to this difference. For example, a founder mutation, c.1534C>T (p.Arg512Cys) in the *PCCB* gene, associated with propionic aciduria, was identified in Nakhon Phanom Province. Additionally, the c.51C>G (p.Phe17Leu) mutation in the *SLC22A5* gene, linked to carnitine uptake deficiency, was found to be more prevalent in the Thai population [10], consistent with its increased allele frequency in East Asian populations, as indicated by the Genome Aggregation Database (gnomAD) with an aggregated allele frequency of 0.001662 [11]. These findings highlight the influence of genetic diversity on the incidence of IEMs and the importance of region-specific genetic studies to tailor screening and second-tier diagnostic strategies.

Maternal conditions affecting NBS results, such as maternal carnitine deficiency, were identified in six cases. These do not count in true-positive cases because it is not the primary objective of NBS.

Among the 20 cases, disorders of organic acid metabolism were the most prevalent, accounting for 7 cases (35%), followed by 6 cases of amino acid metabolism disorders (30%), 4 cases of fatty acid metabolism disorders (20%), and 3 cases of urea cycle disorders (15%). Additionally, the classification of IEMs into urgent and very urgent groups that was initially established by the NHSO of Thailand is not an international standard classification but is instead established based on two criteria: (1) the urgency of following up with patients for evaluation and treatment referral, and (2) the benefits in terms of budget allocation for reimbursement of medical expenses.

All diseases in the very urgent group are characterized by early onset and are life-threatening. Among 20 patients, 8 (40%) were diagnosed with diseases in this category, including 2 cases of citrullinemia type 1 (CIT1), 1 case of ornithine transcarbamylase deficiency (OTC), 2 cases of isovaleric acidemia (IVA), 2 cases of propionic acidemia (PA), and 1 case of methylmalonic acidemia (MAA). All patients diagnosed with CIT1 and OTC died a few days before their NBS results were reported. Citrullinemia type 1 typically presents with severe clinical manifestations and non-favorable outcomes, particularly in the classic type [12]. However, some cases of the non-classic type exhibit milder symptoms with a later onset. Similarly, OTC can have a late onset and milder symptoms in certain cases [13,14]. While expanded NBS may not significantly improve outcomes for the classic type of CIT1, its benefit lies in identifying families at risk and enabling prenatal diagnosis to prevent recurrence in future pregnancies. IVA and PA are among the most common organic acidemias in our region, consistent with the previous report in symptomatic cases [15]. Notably, all PA cases in this study were from the same province, suggesting a possible founder effect, whereas IVA cases were identified in different provinces.

Diseases in the urgent group included those with a later onset and reduced penetrance. Some conditions in this group are considered benign rather than diseases. Among amino acid disorders, tyrosinemia type 3 (TYR3) and hypermethioninemia (MAT) are mostly asymptomatic in early life. TYR3, also known as non-hepatorenal tyrosinemia, is caused by 4-hydroxyphenylpyruvate dioxygenase deficiency. Affected individuals typically exhibit only biochemical changes, including elevated levels of blood tyrosine and massive urinary excretion of its derivatives [16]. However, the long-term neurological outcome in untreated patients remains unclear, and data on the benefits of a strict low-tyrosine diet are limited, making the optimal management of TYR3 controversial [16,17]. Longitudinal follow-up studies are essential for better understanding this condition. In our study, we identified three patients with hypermethioninemia due to methionine adenosyltransferase I/III deficiency (MAT). This is caused by a mutation of the *MAT1A* gene. Although most affected individuals remain asymptomatic, MAT cannot be clearly classified as a benign condition [18]. Some reports suggest neurological complications, including intellectual and learning disabilities [19]. A consensus recommendation suggests that individuals with plasma methionine exceeding 800 μmol/L should be treated with a low-methionine diet [19]. Apart from MAT, homocystinuria (HCY), which is classified as a very urgent disorder, also presents with elevated methionine in NBS. Differentiating both diseases requires additional testing, including homocysteine level and molecular analysis. In our study, no cases of HCY were identified. HCY remains a challenging condition to screen for, as it has a higher false-negative rate due to the naturally low methionine levels in breast milk.

Some organic acid disorders in this panel are considered benign. Isobutyryl-CoA dehydrogenase deficiency (IBD) is one such condition, as most affected individuals are asymptomatic, and evidence regarding long-term neurological deficit remains unclear [20,21]. Similarly, 3-methylglutaconyl-CoA hydratase deficiency (MGA) may also be benign due to potential bias confounding biochemical diagnosis [22]. 2-methylbutyryl-CoA dehydrogenase deficiency (MBD) predominantly found in the Hmong ethnic group is usually asymptomatic [23]. However, MBD shares the same informative marker (C5) with IVA. Therefore, a very urgent recall remains essential for positive C5 cases to ensure no IVA cases are missed.

In fatty acid oxidation defect groups, short-chain acyl-CoA dehydrogenase deficiency (SCAD) is also clearly classified as a benign condition rather than a disease, as it does not require treatment [24]. Reporting SCAD and other benign conditions with an urgent recall level may cause unnecessary parental anxiety. However, in accordance with the national policy, affected infants are still recalled within 48 h, although a hospital visit is not required. Instead, our program allows for evaluation via phone or telemedicine, and we proactively inform parents about the benign nature of this condition to prevent unnecessary concern.

Beyond benign conditions, disorders with low penetrance or a later onset, such as methylcrotonyl-CoA carboxylase deficiency (MCC) and carnitine uptake defect (CUD), are also included in the NBS program [10,25]. However, their identification can be valuable for genetic counseling, allowing families to understand the conditions, be aware of the risks, and be prepared to handle potential symptoms that may manifest later in life.

Including benign conditions and diseases with reduced penetrance in the NBS panel complicates communication between the NBS center and primary hospitals. Regular training of the network is essential to address these challenges effectively. Conversely, some disorders cannot be fully detected through MS/MS-based NBS. Citrullinemia type 2 (CIT2) or neonatal intrahepatic cholestasis caused by citrin deficiency (NICCD) has a low detection rate and a high false-negative rate and requires molecular testing for confirmation [26,27]. Similarly, hyperornithinemia with gyrate atrophy (HOGA) is very difficult to diagnose based on abnormal amino acid levels (elevated ornithine) during the neonatal period. This difficulty arises because, in early life, the ornithine aminotransferase (OAT) reaction may exhibit reversed flux, favoring ornithine synthesis rather than its degradation. As a result, OAT deficiency leads to relatively low ornithine levels, along with deficiencies in its downstream products, arginine and citrulline [28]. Additionally, fatty acid oxidation disorders such as very-long-chain acyl-CoA dehydrogenase deficiency (VLCAD), medium-chain acyl-CoA dehydrogenase deficiency (MCAD), and late-onset multiple acyl-CoA dehydrogenase deficiency (MAD) can yield both false-positive and false-negative results [29,30]. This highlights a critical issue for NBS centers: the need to effectively communicate with primary hospital coordinators and parents regarding potential false positives and negatives, benign conditions, reduced-penetrance diseases, and late-onset disorders to ensure appropriate follow-up and counseling.

Regarding disease classification, there are still areas that require further improvement. National data will provide valuable insights for re-evaluating which diseases should be added or removed from the NBS panel. Collaboration between NBS centers and RD centers, along with the dissemination of accurate and appropriate information, will enhance the overall efficiency and effectiveness of the NBS program nationwide.

In the fiscal year 2024 (1st October 2023, to 30th September 2024), the expanded NBS program for IEMs achieved nationwide coverage in Thailand. Data collected from all RD centers revealed an incidence rate of 1 in 7433 live births among a total of 475,744 newborns. A total of 64 cases were confirmed with a diagnosis of any one of the IEMs. Ten of these cases resulted in mortality, with diagnoses including tyrosinemia type 1 (one case), citrullinemia type 1 (three cases), isovaleric aciduria (one case), carnitine-acylcarnitine translocase deficiency (three cases), very-long-chain acyl-CoA dehydrogenase (VLCAD) deficiency (one case), and multiple acyl-CoA dehydrogenase deficiency (one case). The overall mortality rate across the country was 15.6% of confirmed cases, compared to 8.3% observed in HRs 7 and 8.

A key factor contributing to the program’s success is its strategic location. Srinagarind Hospital, situated within the catchment area, benefits from an efficient consultation and referral system, allowing for quicker and more effective patient management. This advantage enhances timely diagnosis and intervention, ultimately improving patient outcomes. Presenting these outcomes to policy makers could serve as a key driver for policy changes, potentially leading to the establishment of additional RD centers in regions outside Bangkok. Expanding the network of RD centers would enhance access to specialized care, improve diagnostic and treatment efficiency, and ensure equitable healthcare services for patients across the country.

Our study demonstrates the effectiveness of utilizing technology to streamline the entire NBS process, from sample collection to reporting. The use of electronic ordering by 100% of network hospitals significantly reduced human errors and labor hours in the laboratory process. The coordinators at network hospitals can monitor the real-time status of their samples through the KKU-IEM platform, ensuring transparency and efficiency. The program is continuously developed and updated based on user feedback, ensuring it remains user-friendly and responsive to the needs of the coordinators. This collaborative and iterative approach encourages widespread adoption and ease of use among stakeholders. The reduction in the average time from birth to reporting (from 9.13 days in 2023 to 8.4 days in 2024) highlights the program’s commitment to continuous improvement. These enhancements were driven by performance feedback shared at conferences and subsequent workflow adjustments made in collaboration with network hospitals. Faster turnaround times are critical, particularly for conditions such as IEMs, where delays in diagnosis and treatment can result in irreversible developmental delays or even mortality.

Given the program’s strong performance, it has been shared with other NBS centers across Thailand to help improve the overall NBS process nationwide. This initiative not only enhances service quality at a regional level but also contributes to a more efficient national NBS system, ensuring better outcomes for affected infants and their families.

The program’s geographic analysis revealed variations in the duration from birth to report across provinces, with Bueng Kan demonstrating longer times compared to others. This variability could reflect differences in hospital infrastructure, sample transportation logistics, or personnel training levels. Following the implementation of a knowledge management process in 2024, significant improvements were observed in Bueng Kan Province. Adjustments made to the processes in network hospitals resulted in a notable 19.1% reduction in reporting time. This improvement highlights the effectiveness of data-driven interventions and collaborative problem-solving in enhancing service delivery and ensuring more timely care for affected newborns.

The success of the expanded NBS program in rural Thailand emphasizes the feasibility of implementing such initiatives in resource-limited settings. Integrating digital tools, such as the KKU-IEM online platform, played a critical role in streamlining processes from registration to result reporting. Collaboration with TPD for sample logistics ensured timely and efficient transportation, even in remote areas. These innovations can serve as a model for other developing countries seeking to establish or scale up their own NBS programs.

Our NBS program also benefited from a robust quality control system, including participation in international proficiency testing and inter-laboratory comparisons. These measures not only ensured high standards but also aligned the program with global benchmarks, bolstering its credibility and sustainability.

Moving forward, the program should focus on further reducing reporting times, particularly for very urgent conditions, by utilizing real-time data analytics and automated result reporting systems. Expanding public education campaigns on the importance of NBS and involving community health workers in follow-up care could enhance parental compliance and reduce missed follow-ups.

Additionally, conducting long-term outcome studies on affected infants would provide critical insights into the efficacy of early interventions and help shape future policy decisions. Another important step would be to develop second-tier tests for positive results, enhancing diagnostic specificity. The program should also consider including additional diseases that pose significant regional challenges but are preventable or treatable, further broadening its impact.

## 5. Conclusions

The expanded NBS program in HRs 7 and 8 of Thailand presents a successful model for rural implementation, achieving 98.6% coverage and reducing reporting times to 8.4 days in 2024. The integration of the KKU-IEM platform and collaboration with network hospitals ensured operational efficiency and timely management. This model emphasizes the value of data-driven improvements, region-specific strategies, and comprehensive follow-up systems in enhancing NBS programs, serving as a model for similar initiatives in resource-limited settings.

## Figures and Tables

**Figure 1 IJNS-11-00026-f001:**
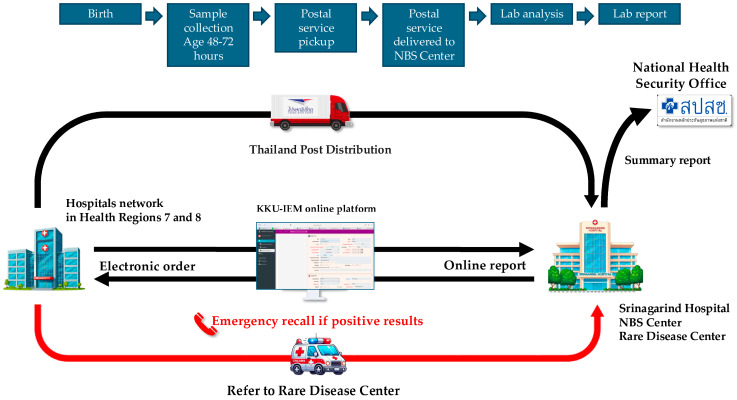
The workflow system of expanded NBS in Health Regions 7 and 8.

**Figure 2 IJNS-11-00026-f002:**
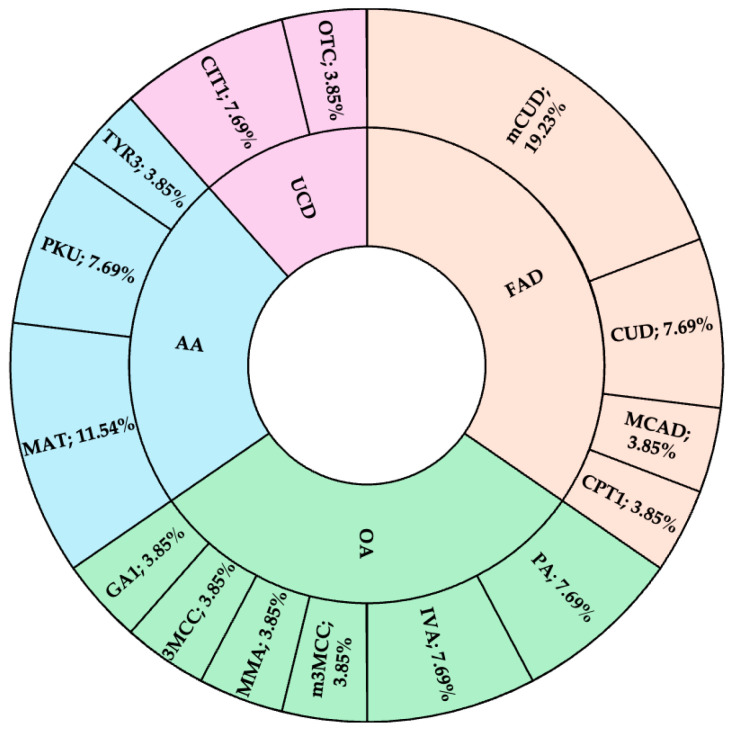
All IEMs diagnosed by this NBS program. AA: disorders of amino acid metabolism; CPT1: carnitine palmitoyltransferase type 1 deficiency; CUD: carnitine uptake defect; CIT1: citrullinemia type 1; FAD: disorders of fatty acid oxidation; GA1: glutaric acidemia type 1; IVA: isovaleric acidemia; MAT: hypermethioninemia (methionine adenosyltransferase deficiency); 3MCC: 3-methylcrotonyl-CoA carboxylase deficiency; MMA: methylmalonic acidemia; m3MCC: maternal 3-methylcrotonyl-CoA carboxylase deficiency; mCUD: maternal carnitine uptake defect; MCAD: medium-chain acyl-CoA dehydrogenase deficiency; OA: disorders of organic acid metabolism; OTC: ornithine transcarbamylase deficiency; PA: propionic acidemia; PKU: phenylketonuria; TYR3: tyrosinemia type 3; UCD: urea cycle disorder.

**Table 1 IJNS-11-00026-t001:** Forty target disorders for IEMs in expanded NBS and classification according to the urgency level.

Groups	Target Diseases	Abbreviation	Urgency Levels *
Disorders of amino acid metabolism	1. Phenylketonuria	PKU	Urgent
	2. Tetrahydrobiopterin defects	BH4	Urgent
	3. Maple syrup urine disease	MSUD	Very Urgent
	4. Tyrosinemia type 1	TYR1	Very Urgent
	5. Tyrosinemia type 2	TYR2	Urgent
	6. Tyrosinemia type 3	TYR3	Urgent
	7. Homocystinuria	HCY	Urgent
	8. Hypermethioninemia	MAT	Urgent
	9. Hyperornithinemia with gyrate atrophy	HOGA	Urgent
Disorders of organic acid metabolism	1. Glutaric acidemia type 1	GA1	Urgent
	2. Isovaleric acidemia	IVA	Very Urgent
	3. Methylmalonic acidemia	MMA	Very Urgent
	4. Propionic acidemia	PA	Very Urgent
	5. Multiple carboxylase deficiency	MCD	Very Urgent
	6. Adenosylcobalamin synthesis defects	Cbl A/B	Very Urgent
	7. Beta-Ketothiolase deficiency	BKT	Very Urgent
	8. 3-Hydroxy-3-Methylglutaryl-CoA lyase deficiency	HMG	Very Urgent
	9. Isobutyryl-CoA dehydrogenase deficiency	IBD	Urgent
	10. 2-Methylbutyryl-CoA dehydrogenase deficiency	MBD	Very Urgent
	11. 3-Methylcrotonyl-CoA carboxylase deficiency	MCC	Very Urgent
	12. 3-Methylglutaconyl-CoA hydratase deficiency	MGA	Very Urgent
	13. Malonic aciduria	MA	Urgent
	14. Combined methylmalonic acidemia and homocystinuria	Cbl C/D	Very Urgent
Urea cycle disorders	1. Citrullinemia type 1	CIT1	Very Urgent
	2. Citrullinemia type 2 or Citrin deficiency	CIT2	Very Urgent
	3. Argininosuccinic aciduria	ASA	Very Urgent
	4. Argininemia	ARG	Urgent
	5. Hyperammonemia-Hyperornithinemia-Homocitrullinuria syndrome	HHH	Urgent
	6. Ornithine transcarbamylase deficiency	OTC	Very Urgent
Disorders of fatty acid oxidation	1. Medium-chain acyl-CoA dehydrogenase deficiency	MCAD	Urgent
	2. Long-chain hydroxyacyl-CoA dehydrogenase deficiency	LCHAD	Very Urgent
	3. Very-long-chain acyl-CoA dehydrogenase deficiency	VLCAD	Very Urgent
	4. Short-chain acyl-CoA dehydrogenase deficiency	SCAD	Urgent
	5. Short-chain hydroxyacyl-CoA dehydrogenase deficiency	SCHAD	Urgent
	6. Trifunctional protein deficiency	TFP	Very Urgent
	7. Multiple acyl-CoA dehydrogenase deficiency	MAD	Very Urgent
	8. Carnitine-acylcarnitine translocase deficiency	CACT	Very Urgent
	9. Carnitine palmitoyltransferase type 1 deficiency	CPT1	Urgent
	10. Carnitine palmitoyltransferase type 2 deficiency	CPT2	Very Urgent
	11. Primary systemic carnitine deficiency (carnitine uptake defect)	CUD	Urgent

* Urgency levels: very urgent (the baby must be followed up for evaluation within 24 h after receiving the positive results) and urgent (the baby must be followed up for evaluation within 48 h after receiving the positive results).

**Table 2 IJNS-11-00026-t002:** Summary of NBS outcomes, confirmatory testing, and incidence rates. IEMs: inborn errors of metabolism; CH: congenital hypothyroidism.

Tests	Newborn Screening (Neonates)	Positive Results (Cases)	Recall Rate (%)	Number of Infants Undergoing Confirmatory Tests (%)	Confirmed Cases	Incidence Rate	Positive Predictive Value (%)
CH	122,004	287	1:425.1 (0.24)	284 (99.0)	101	1:1208 (0.08)	35.2
IEMs	122,004	529	1:230.6 (0.43)	508 (96.0)	20	1:6100 (0.02)	3.8

## Data Availability

The data are contained within the article.

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
