# Peer review of "The Establishment of Expanded Newborn Screening in Rural Areas of a Developing Country: A Model from Health Regions 7 and 8 in Thailand"

_2409-515X, 2025, doi:10.3390/ijns11020026_

Round 1

Reviewer 1 Report

Comments and Suggestions for Authors

Dear authors, thank you for the chance to review your manuscript,

Congratulation to the implementation of this expanded NBS in HR7 and 8. I think your implementation of electronic ordering and reporting is the most interesting part of your work. Besides of course the reported incidence rates are also interesting, as are the details on genetic background. However, there are still a few major concerns I have, that need to be adressed.

  1. The basis for the introduction of IEM into the NBS panel seems to be made on the basis of ref. 4. However, I cannot find this publication. It is not in PubMed, and on the journals homepage vol. 18 is not published in 2024
  2. If the authors include urgancy and very high urgency, then this should be substantiated by literature references. 
  3. In addition there several disorders labelled , which a widely classified as non-disease, or at least as condicions with low disease penetrance. i.e.: SCAD, 3-MCC, IBD, MBD
  4. In other disorders like citrullinaemia type 1, NBS mostly is too late, since babies often decompensate already during the first days of life.
  5. And at least for 2 of the listed disorders, HOGA, and Homocystinuria, it is doubtful whether you can find these cases by amino acids during the newborn periode.

Without a decent discussion about the disorders, the presentation is missleading.

Reviewer 2 Report

Comments and Suggestions for Authors

Summary

This report describes the expansion of newborn screening (NBS) in two Health Regions in Thailand to include screening for congenital hypothyroidism (CH) and 20 inborn errors of metabolism (IEMs).  CH screening is done by measuring thyroid stimulating hormone (TSH) with a valid and standardized assay and IEMs are detected with tandem mass spectrometry.  It describes the processes, the outcomes, how the experience compares to and might be educational for other regions.

Strengths

The manuscript provides a thorough description and analysis of the processes of implementing the expansion and a clear accounting of the results of the screening.  The discussion makes appropriate comparisons to other programs in Thailand and the rest of the world.  It also suggests areas for improvement.  Figure 3 provides a nice depiction of the distribution of IEMs that I’ve not seen previously.

Areas for Improvement of the manuscript

In the first paragraph of Results, the numbers of preterm and low birth weight newborns needs to be clarified to remove the ambiguity that they might be additive.  For example, “Among them nnn were preterm (12,164) and/or low birth weight (13,179).”  The nnn is likely 14,000-15,000 as there is a great deal of overlap between preterm and low birthweight.

I am fairly sure, but not certain, that IEM positive screens in babies born to mothers with IEMs are considered false positive results as the vast majority of IEMs are autosomal recessive.  If that is correct, they should not be counted as true positives as they have been in a few places in the manuscript and line 310 needs to be changes or removed.  Likewise, most do not consider family screening (lines 312-313) to be one of the goals of NBS.

Minor Points

There are many places (lines 165, 166, 265, 266, 270 and maybe others) where CH, the disease, is referred to as TSH.

In Table 1, under the Urgency column, diseases should be characterized as “Urgent” or “Very Urgent”, not “Urgency” and “Very Urgency”.  There are numerous other minor misuses of English that the editors can correct.

A potential cause of transient CH that might be worth mentioning in the Discussion is iodine deficiency.  If iodine deficiency has a high prevalence in the studied areas of Thailand that might explain why the birth prevalence of CH is so high.

I don’t think Figures 2 or 4 are needed.  The precise geography of NBS results does not add to the value of the manuscript.

I think that the paragraph that starts on line 193 belongs in the Discussion and not in the Results.  It reports NBS findings in the whole country, which is not the focus of this report.

The first sentence on line 251 is redundant, as is line 126.

The PPVs reported on line271 should be included in the Results.

Comments on the Quality of English Language

The manuscript is nicely organized and written, but needs correction of some English language usage.  One specific instance is cited in Suggestions for Authors but there are several others.

Round 2

Reviewer 1 Report

Comments and Suggestions for Authors

Dear authors,

thank you for the revised version of the manuscript. All open questions and remarks have been address and the  manuscript has been revised accordingly.

Best regards,